# The Evolution of the Tensile Properties of MoS_2_-Coated Titanium Alloy Bolts Under the Synergistic Damage of NaCl Corrosion and Preloading

**DOI:** 10.3390/ma18010123

**Published:** 2024-12-31

**Authors:** Derong Feng, Maoyang Xie, Weilin Yu, Chao Li, Raolong Guo, Yunpeng Hu, Quanyuan Ming, Qiang Wan

**Affiliations:** 1Henan Aerospace Precision Manufacturing Co., Ltd., Xinyang 464000, China; xyfdrong@126.com (D.F.); xiemy369@163.com (M.X.); yuweilin.hi@163.com (W.Y.); chaogto@163.com (C.L.); 13383768531@163.com (R.G.); hyp1378293@163.com (Y.H.); 2Henan Key Laboratory of Fastening and Connection Technology, Xinyang 464000, China; 3College of Engineering, Huazhong Agricultural University, Wuhan 430070, China; mingquanyuan1023@163.com

**Keywords:** assembly TC4 bolt, MoS_2_ coating, anodic oxidation, corrosion, tensile property

## Abstract

MoS_2_ coating is a newly developed method to prevent bolt corrosion and the seizure of bolts used in equipment in sea areas. It is of great significance to investigate the evolution of the tensile properties and intact coatings for the maintenance of coated bolts. To evaluate the tensile properties of MoS_2_-coated titanium alloy bolts, titanium alloy bolts coated with MoS_2_ (TC4+MoS_2_) and bolts treated with a composite treatment of anodizing oxidation and MoS_2_ coating (TC4+AO+MoS_2_) were corroded in salt spray tests for 4300 h. The MoS_2_ coating significantly enhanced the bolts’ corrosion resistance, demonstrating exceptional protective performance by only experiencing minor peeling due to oxidation-induced cracking of the coating during the extensive 4300 h salt spray test. The tensile strengths of the TC4+MoS_2_ and TC4+AO+MoS_2_ bolts both decreased as compared with the original bolts. The bolts pretreated with anodic oxidation revealed lighter coating peeling and maintained a higher tensile strength after corrosion. Therefore, it can be concluded that the coatings provided excellent corrosion resistance, leading to a minor impact on the bolts’ tensile strength and fracture behavior under the synergistic damage of sea water corrosion and preloading.

## 1. Introduction

TC4 (Ti6Al4V) bolts are widely used in aerospace, shipbuilding, biomedical, and other fields due to their excellent mechanical properties, corrosion resistance, and low density [1,2]. Seizing is an important failure mode in the application of titanium alloy bolts [3]. The fundamental reason for seizing is enhanced galling, which is a form of wear caused by metal surfaces in gliding contact with one another. Enhanced galling is attributed to increased friction [4]. There are three reasons for the increased wear of titanium bolts. First, the high stress induced by preloading greatly increases the friction coefficient [5]. Second, the smooth surface brings about a large contact area between threads, which intensifies the friction [6]. Third, a lack of lubrication can also lead to an increase in the friction coefficient [6]. Currently, coatings with a low friction coefficient are being developed to avoid the increased wear and prevent seizing [7,8].

The coatings used for bolts mainly include hot-dip galvanizing, zinc–nickel alloy coating, Dacromet coating, and fluorocarbon coating [9]. However, hot-dip galvanizing coatings with poor corrosion resistance and a high friction coefficient of over 0.15 are not suitable for applications in titanium bolts. Dacromet coating and fluorocarbon coating, with a friction coefficient of 0.08–0.12, present poor adhesion with the bolt substrate and are limited as they providing a low friction coefficient over a long duration. Also, another limiting factor in the application of the above two coatings is aging failure. Obvious exfoliation was observed after 1200 h of corrosion in a salt spray corrosion test.

Molybdenum disulfide (MoS_2_) is a layered structure that permits the easily sliding of layers under the action of shear force, thereby significantly reducing the friction coefficient [10]. Meanwhile, it possesses good corrosion resistance with doping metal particles and multi-functional structures [11,12]. Therefore, it has been widely used to coat titanium alloy bolts in recent years [13]. TC4 bolts coated with molybdenum disulfide show excellent tribological properties, which can effectively reduce the friction and wear of bolts. Under the same pre-tightening torque conditions, molybdenum-disulfide-coated bolts present the best anti-loosening performance due to their high pre-tightening force [14] and are currently the most promising bolt protective coating [15].

The main problem of MoS_2_ coating is the peeling of the coating [16,17,18], especially under preloading. The stress resulting from preloading can accelerate crack propagation in the coating and the passivation film formed during corrosion, accelerating the corrosion of the bolt substrate. Further improving the coating’s adhesion and ensuring the service life of the coating is key to ensuring the long-term effectiveness of the coating. Pulse anodizing technology can form micropores on the surface of titanium alloy, which, in turn, enhances the adhesion of coatings by providing binding sites [19,20]. The combination of pulse anodizing and MoS_2_ coating has become a new strategy in the surface treatment of TC4 bolts.

Many studies on the friction properties and anti-seizing properties of MoS_2_ coating on bolts have been carried out. Unlike fluid lubrication, MoS_2_ is suitable for use in vertically arranged components because it lacks migration. For this reason, sputtered MoS_2_ coating is primarily considered as a low-friction/anti-seizing coating method of the ITER blanket central bolt insert [21]. Zhu attributed the best anti-loosening performance to MoS_2_ coating on a bolt compared to PTFE and TiN coatings [15,22]. Usually, titanium alloy bolts are subjected to NaCl corrosion during service. Corrosion can induce the failure of MoS_2_ coatings and can even lead to the corrosion of the substrate when the coating fails [23,24]. This will greatly decrease the tensile bearing capacity of the bolt under the action of salt spraying and tension [25,26]. Wei et al. revealed that the ultimate tensile strength and load-bearing capacity of M24 × 130 mm 20 MnTiB bolts corroded after 10 years in air decreased by 2.5–6.8% and 1.25–5.5%, respectively [27]. Notably, the load at 75% displacement dropped from 390 kN to 340 kN, a reduction of over 15%. Corrosion reduces the clamping force, slip coefficient, initial slip load (or friction shear strength), full slip load, yield load, ultimate load, and ductility, all of which are closely related to the corrosion loss of the entire specimen [28]. The slip resistance coefficient (*μ*) of specimens after corrosion is approximately linear with respect to the increase in the steel mass loss rate [29]. Therefore, exploring the mechanical property evolution and coating stability of TC4 under the synergistic action of preloading and salt spraying to evaluate the service duration of the surface coating is key to the application of coated bolts used in hot and humid marine environments.

In this study, corrosion tests were carried out on titanium alloy bolts coated with MoS_2_ and bolts treated through a composite treatment of anodizing and MoS_2_ coating via a salt spray test and coastal hanging corrosion for 4300 h. After corrosion, the bearing capacity of the bolts was tested via a tensile test. The surface morphology and tensile fracture morphology of the bolts were observed in this study in order to explore the underlying mechanisms for the failure of the coating in the salt spray environment and to confirm the protective aging effect of the MoS_2_ coating on the bolts.

## 2. Materials and Methods

### 2.1. Corrosion Tests

M12-TC4 bolts, provided by Henan Aerospace Precision Manufacturing Co., Ltd., Xinyang, China, were either coated with MoS_2_ or treated with a composite treatment of anodizing and MoS_2_ coating, and the samples were named TC4+MoS_2_ and TC4+AO+MoS_2_. The MoS_2_ coating process is as follows: first, the bolts were degreased and then cleaned in hot water, acid, and hot pure water in order; next, the bolts were dried and coated with MoS_2_ via immersion; third, the MoS_2_ was allowed to solidify. The bolts were installed in an aluminum fixture with a pre-tightening force of 40 kN, which is the installation requirement for M12 bolts in applications. The corrosion test of the coated bolts was carried out using a salt spray corrosion test. The salt spray corrosion test was conducted in a salt spray corrosion test chamber according to the GB/T 10125-2021 standard. During the test, a 3.5% NaCl neutral aqueous solution with a pH value between 6.5 and 7.2 was used as the corrosive solution at 35 °C. During the test, the uniformity and stability of the spray were strictly controlled to ensure the consistency and repeatability of the corrosive environment. The salt spray corrosion times were set at 96, 192, 1100, 2100, and 4300 h. The times for the coastal hanging corrosion test were 2100 h and 4300 h.

### 2.2. Friction Tests

The friction coefficient of the MoS_2_-coated TC4 plates with and without pulse anodic oxidation treatment was tested by a sliding friction tester (RTEC MFT-5000). Plate specimens with a diameter of 20 mm were polished to a mirror finish. The thread friction coefficient test was conducted on the coated bolts in accordance with the ISO 16047-2005 “fasteners—torque/clamp force testing” standard after the salt spray corrosion, as shown in Figure 1. The axial force of the bolt was set at 40 kN. The experiment was performed on a SCHATZ threaded fastener analysis system (5416-2777/04/CS). Each set of experiments was repeated three times, and the average value of the friction coefficients was taken as the final result.

### 2.3. Tensile Test of Assembly Bolts After Corrosion

The tensile testing was conducted on multiple pairs of assembled bolts using a CMT5305 electronic universal testing machine in accordance with the Chinese mechanical testing standards for fasteners, specifically GB/T 3098.1-2010. The stress and elongation were recorded for each test. Three specimens were tested for each corrosion time to obtain the average values of the yield strength and tensile strength. The yield strength was determined at the turning point between linear and nonlinear changes from the force–displacement curves. The tensile strength was calculated from the maximum force of the tensile curve by dividing by the area of the fracture cross-section. Bolts exposed at the Sanya coastal area for 2100 and 4300 h were also tensioned to identify the effectiveness of the salt spray corrosion.

### 2.4. Microstructure Characterization

The microstructure and chemical composition of the treated bolts were characterized with an FEI scanning electron microscope (SEM) system equipped with energy-dispersive spectrometry (EDS) at the surface and cross-section. Additionally, the surface morphology of the corroded specimens and the fracture surfaces of the tensile failures were characterized and analyzed to identify the corrosion products and fracture morphology. Specimens for cross-sectional observation were cut from the bolts. Then, they were ground and polished to a mirror-like surface. The characterization of the corroded MoS_2_ coating and the fracture morphology was conducted from the original state. The working distance of the SEM was 13 mm, and the working voltage was 20 kV.

## 3. Results and Discussion

### 3.1. Microstructure of the Coating on Bolts

Figure 2 shows the cross-sectional morphology and compositions of the MoS_2_-coated bolts. EDS mapping profiles and line scans were carried out to reveal the elemental distribution in the coatings. According to the elemental distribution of the mapping profiles, the upper layer was mainly composed of Mo and S, while the lower layer was rich in Ti, V, Al, and O. The results suggested that the upper layer was the MoS_2_ coating, and the lower layer was the TC4 substrate. The line scanning distribution also confirmed the above conclusion. The width of the Mo and S peaks in the element scan lines along the cross-section was the thickness of the MoS_2_ coating. Combined with the thickness calculated from the morphology figures, the MoS_2_ coating was about 42.55 μm thick. The observed O element was attributed to pollution during the specimen production process. Figure 3 shows the cross-sectional morphology and compositions of the TC4+AO+MoS_2_ bolt. Similarly, the top MoS_2_ coating and the TC4 substrate were confirmed by the aggregation of S and Ti in the top surface and the bottom layer. An obvious O peak was found between the MoS_2_ coating and the TC4 substrate. According to the process, this should be the anodic oxidation layer, which mainly consisted of Ti and O.

### 3.2. Friction Coefficient and Thread Friction Coefficient Test

Figure 4 reveals the friction curves of the plates treated with the MoS_2_ coating and with the composite treatment of anodic oxidation and MoS_2_ coating. The friction coefficient (COF) of the plate coated with MoS_2_ only increased with the slide time and reached a maximum value. The average friction coefficient was calculated to be 0.166. In comparison, the friction coefficient of the plate treated with the composite treatment of anodic oxidation and MoS_2_ coating revealed a similar trend but had a higher average friction coefficient of 0.192. The higher friction coefficient was related to the higher roughness induced by anodic oxidation [30].

The thread friction coefficient of the two treated bolts was calculated by measuring the tightening torque and thread torque according to Formulas (1) and (2).
(1)$$\mu_{tot}=\frac{\frac{T}{F}-\frac{P}{2\pi }}{0.5777d_{2}+0.5D_{b}}$$


(2)
$$T=T_{b}+T_{th}$$


μ_tot_ is the thread friction coefficient, d_2_ is the pitch diameter, d_b_ is the friction diameter of the nut bearing surface, p is the pitch, t is the tightening torque, t_b_ is the torque of the end-face friction, T_th_ is thread torque, and f is the axial force.

The thread friction coefficient of the bolt coated with MoS_2_ was 0.062, while that of the TC4+AO+MoS_2_ bolt was 0.065. The thread friction coefficient was consistent with the sliding friction coefficient obtained from plates [31]. However, the friction coefficient of the bearing surface of the bolt coated only with MoS_2_ was 0.056, and that of the bolt additionally treated with anodic oxidation was 0.049. A lower friction coefficient could lead to more tightening torque being converted into the axial preloading of the bolt and decrease the wear during assembly or disassembly [32]. Also, the lower friction coefficient of the bearing surface of the TC4+AO+MoS_2_ bolt resulted in a slightly smaller total friction coefficient of 0.055.

### 3.3. Surface Morphologies of Corroded Bolts

Figure 5 and Figure 6 show the macroscopic morphology of the bolts after surface passivation and molybdenum disulfide coating under long-term 3.5% NaCl salt solution spraying. The salt spray corrosion times were 0 h, 192 h, 1100 h, 2100 h, and 4300 h. As can be seen from the figures, after 4300 h of salt spray corrosion, all the bolts basically showed no macroscopic rust spots, indicating good corrosion resistance. However, the coating on both the bolt head and the screw rod had peeled off, exposing the TC4 bolt substrate. Among these, the peeling area ratio of the bolt head was less than that of the screw rod. The reason is that during the bolt removal process, the screw rod rubbed against the clamping nut, causing the surface coating to peel off. As the corrosion time increased, the peeling area of the coating on the screw rod surface of the bolts treated with the two methods increased significantly. It can be seen that the corrosion seriously reduced the stability of the coating, and there was no significant difference in the peeling area between the screw rod of the bolt with anodic oxidation treatment and the screw rod without anodic oxidation treatment.

Figure 7, Figure 8 and Figure 9 show the microscopic morphologies and compositions of the TC4 bolts coated with MoS_2_ coating and with the composite treatment of anodic oxidation and MoS_2_ coating after the corrosion test. Figure 7 shows that significant coating delamination occurred on the surface of the MoS_2_-coated bolt after 192 h of salt spray corrosion. The EDS mapping profiles indicated that the exposed areas were primarily composed of Ti, Al, and V elements, which are the main components of the TC4 substrate. It is suggested that delamination of the coating occurred. The surface layer mainly consisted of Mo and S elements, indicating that the surface coating was MoS_2_. A high density of O elements about 1.6 at. % was observed on the specimen surface in the mapping profiles, indicating oxidation of the MoS_2_. Due to the delamination of the surface coating, the TC4 alloy substrate also oxidized with a 15.3% O content. This suggests that the delamination of the MoS_2_ led to a loss of corrosion protection. As the salt spray corrosion time increased, the O content in the MoS_2_ coating further increased [23,33]. For the specimen corroded for 2100 h, the O content in the MoS_2_ coating increased to 22.2%. Correspondingly, the TC4 substrate material also underwent more severe corrosion, with the surface O content increasing to over 44%. After 4300 h, the O content in the coating surface further increased to 44.6%, accompanied by significant cracks on the coating surface. However, the O content in the oxide layer on the TC4 surface did not increase further. The results indicated that the MoS_2_ coating oxidized in the salt spray corrosion environment, leading to the cracking and subsequent delamination of the coating due to the stress growth during the oxidation process, which resulted in the exposed TC4 substrate undergoing metal corrosion.

The surface of the bolts with the composite treatment showed a uniform distribution of Mo and S. No obvious presence of Ti, Al, or V elements was observed after 192 h of salt spray corrosion. The results indicated that there was no MoS_2_ delamination on the surface of the bolts. The EDS spectrum analysis revealed that the surface oxygen content was consistently around 29%, which was higher than that of the bolts without anodic oxidation treatment. However, the surface MoS_2_ coating on the bolts treated with pulse anodic oxidation was more complete, even after severe oxidation. This suggested that the anodic oxidation enhanced the adhesion of the coating in the salt spray environment, ensuring the effective protective performance of the MoS_2_ coating [19]. As the salt spray corrosion test was prolonged to 2100 h, the coating on the bolt surface started to exfoliate, with the presentation of Ti elements, as shown in Figure 8. Spot analysis revealed that the oxygen content in the exposed substrate surface reached 45%, which was significantly higher than the 22.2% observed on the corroded substrate of the MoS_2_-coated bolts. The high oxygen content was not derived from corrosion of the exposed substrate but from the oxide layer formed by anodic oxidation. Since the MoS_2_ coating had not completely exfoliated, the measured oxygen content was lower than that in the anodic oxide layer. Therefore, when the corrosion time was increased to 4300 h, more of the MoS_2_ coating delaminated, and the measured oxygen content on the surface of the bolt substrate further increased to 59.85%.

It is evident that these MoS_2_ and anodic oxidation treatments can improve the corrosion resistance of bolts, but the bonding strength between the coating and the substrate may decrease during long-term exposure to corrosive environments. A decrease in the bonding strength can lead to easy exfoliation of the coating, exposing the substrate material and further exacerbating corrosion [34]. Pulsed anodic oxidation can improve the adhesion of the coating, slowing down the process of coating exfoliation during the oxidation of MoS_2_.

### 3.4. Tensile Properties of Bolts After Corrosion

Figure 10 shows the tension curves of the two different types of treated bolts after salt spray corrosion. The yield strength and tensile strength are shown in Table 1 and Table 2. The tension process indicates that the bolts underwent two stages: elastic deformation and plastic deformation. During the elastic deformation stage, the load on the bolt increased linearly with increasing displacement. When the load reached the yield strength of the material, the curve began to deviate from linearity and entered the plastic deformation stage. Further increasing the load, the tensile strength point was reached. Then, local necking appeared, resulting in a reduction in the actual load-bearing area and a rapid decrease in the load. According to the standard for the mechanical testing of fasteners (ISO 898-1:2013, fitting formulas were developed on the basis of the line segment corresponding to the elastic deformation of the bolts. The yield strength and tensile strength of the bolts after corrosion were obtained.

Figure 11 shows tensile curves of the TC4+MoS_2_ and TC4+AO+MoS_2_ bolts after different times of coastal hanging corrosion, respectively. Figure 12 shows the yield strength and tensile strength of TC4+MoS_2_ and TC4+AO+MoS_2_ bolts after different times of salt spray corrosion and coastal hanging corrosion. When the corrosion time was shorter than 1100 h, both the yield strength and tensile strength exhibited slight fluctuations within a certain range without a significant decrease. This indicated that short-term corrosion revealed little impact on the tensile mechanical properties of the MoS_2_-protected bolts. When the corrosion time increased from 1100 h to 2100 h, the yield strength of both types of bolts showed a significant decrease, dropping from above 1200 MPa to around 1150 MPa. With the further increase in the spray corrosion time, the yield strength of the bolts continued to decrease, reaching 1000 MPa after 4300 h of corrosion. The tensile strength of both types of bolts also followed a similar trend. Within the first 1100 h of corrosion, there was almost no impact on the tensile strength of the bolts. As the corrosion time increased to over 2100 h, the tensile strength of the bolts decreased. After 4300 h, the tensile strengths of the TC4+MoS_2_ and TC4+AO+MoS_2_ bolts were 1332 and 1344 MPa, respectively. The decrements were 38 and 26 MPa compared with those of the original bolts. The TC4+AO+MoS_2_ bolt revealed a smaller decrement compared with the TC4+MoS_2_ bolt. The main reason for the decreased strength was the increased number of crack sources, the reduced effective bearing area, and the shortened crack propagation path of the bolt, resulting from the surface corrosion layer of the TC4 substrate [28]. Under the synergistic effect of preloading and corrosion, the corrosion sites on the surface became the weak points of the bolt, which were prone to cracking, thereby increasing the number of crack sources and decreasing the bearing area [35]. As the corrosion time increased, more corrosion sites formed and more crack sources appeared, leading to a continuous decrease in the tensile strength. Furthermore, the bolts hanging in the coastal area revealed a higher tensile strength as compared with those subjected to the salt spray test after the same corrosion time. This indicated that the damage caused by salt spraying was more severe than that caused by hanging in coastal areas. In addition, it should be pointed out that the tensile strength of the bolts treated with pulse anodic oxidation was higher than that of the untreated bolts under different corrosion times. It can be seen that the pulse anodic oxidation could reduce the tensile strength degradation of the bolts under preloading and corrosion damage by enhancing the adhesion of the MoS_2_ coating and ensuring the long-term anti-corrosion protection of the MoS_2_ coating. Therefore, the composite treatment of anodic oxidation and MoS_2_ coating is a better method for protecting TC4 bolts in corrosion environments.

### 3.5. Fracture Morphology

The fracture morphology was observed by SEM, revealing the fracture mechanism of the corroded bolts, as shown in Figure 13 and Figure 14. After the corrosion for various durations, the fracture mechanism of TC4 bolts coated with MoS_2_ and with the composite treatment of anodic oxidation and MoS_2_ coating was still ductile fracturing, featuring prominent propagation zones and shear lips [36]. The cracks originated from the root of the threads, which are marked by red dashed lines and are referred to as the “initiation zone” [37]. The cracks extended linearly from one side of the thread root to the other, typically situated at stress concentration points or material defects such as corrosion sites. High-magnification observations were conducted on the crack propagation zones marked by the red dashed lines and the shear lips marked by yellow dashed lines. At the microscale, distinct dimples were observed on the bolt fracture surfaces. These dimples were tiny voids formed during the plastic deformation process under tension. As the stress increased, these voids gradually expanded and coalesced, ultimately leading to fracturing. The large number of dimples indicated that the bolts’ tensile fracture process was ductile fracturing. Furthermore, strips could be discerned around the dimples, representing the trajectories of the crack propagation, which indicated the direction of expansion of the cracks after their initiation, as illustrated by the yellow dashed line icons in the figures [38].

## 4. Conclusions

A 4300 h salt spray test and a 4300 h coastal hanging corrosion test were carried out on TC4 bolts coated with MoS_2_ coating and with a composite treatment of anodic oxidation and MoS_2_ coating. By combining the results of reciprocating friction tests and thread friction coefficient measurements, the friction performance of the bolts was investigated. Additionally, the tensile properties and the morphology before and after corrosion were tested. The following conclusions were drawn:

(1) The thickness of the coating on the treated bolt surface was about 42 μm. The reciprocating dry friction coefficients for the MoS_2_ coating application and anodic oxidation + MoS_2_ coating treatment were 0.166 and 0.192, respectively, and the corresponding friction coefficients of the treated bolt threads were 0.062 and 0.065.

(2) After 4300 h of salt spray corrosion, all the bolts basically did not show any macroscopic rust spots, demonstrating good corrosion resistance resulting from the MoS_2_ coating. However, the coating on the bolt surface oxidized, resulting in the peeling of the coating on both the bolt head and the screw rod, which caused the base titanium alloy to oxidize. As the corrosion time increased, the peeling area of the coating of the screw rods of both treated bolts increased significantly. Pulsed anodic oxidation could improve the adhesion of the coating, slowing down the peeling process of the coating during oxidation by revealing a smaller area of peeling off.

(3) When the corrosion time was less than 1100 h, the yield strength and tensile strength fluctuated within a certain range without significant reduction. After 4300 h of corrosion, the decrements in the tensile strength of the TC4+MoS_2_ and TC4+AO+MoS_2_ bolts were 38 and 26 MPa. The bolts treated with pulsed anodic oxidation had higher tensile strength than those without pulsed anodic oxidation under the same corrosion times. Fracture analysis showed that under the combined effect of corrosion and preloading, the bolt fracture mechanism was still ductile fracturing.

The research results showed that MoS_2_ coating is a good means of bolt corrosion protection, and it can effectively reduce the friction coefficient of the bolt surface. Meanwhile, anodic oxidation treatment can improve the long-term effectiveness of MoS_2_ coating protection. Therefore, the composite treatment of anodic oxidation + MoS_2_ coating is an effective means of preventing the corrosion and seizing of titanium alloy bolts. The results provide scientific data for the maintenance of coated bolts. Longer corrosion tests and friction behavior tests should be further conducted to reveal the effectiveness of MoS_2_ coating in protecting bolts from fracturing and wear.

## Figures and Tables

**Figure 1 materials-18-00123-f001:**
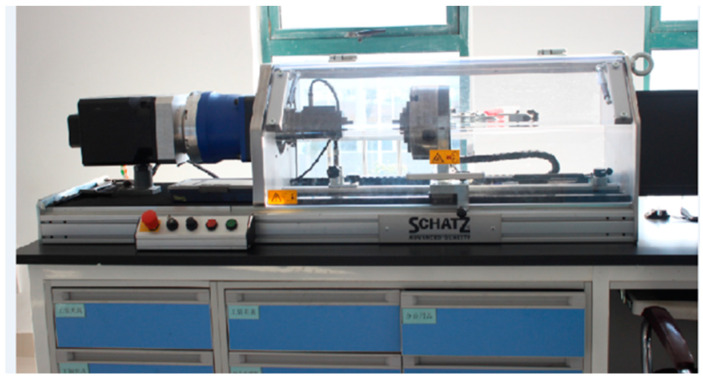
SCHATZ threaded fastener analysis system.

**Figure 2 materials-18-00123-f002:**
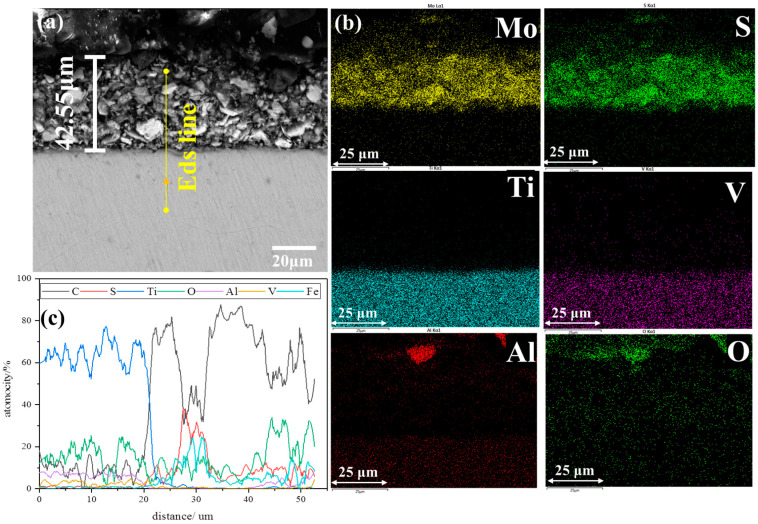
Cross-sectional morphology of TC4+MoS_2_ bolts: (**a**) cross-sectional morphology, (**b**) EDS mapping profiles, (**c**) EDS line scanning.

**Figure 3 materials-18-00123-f003:**
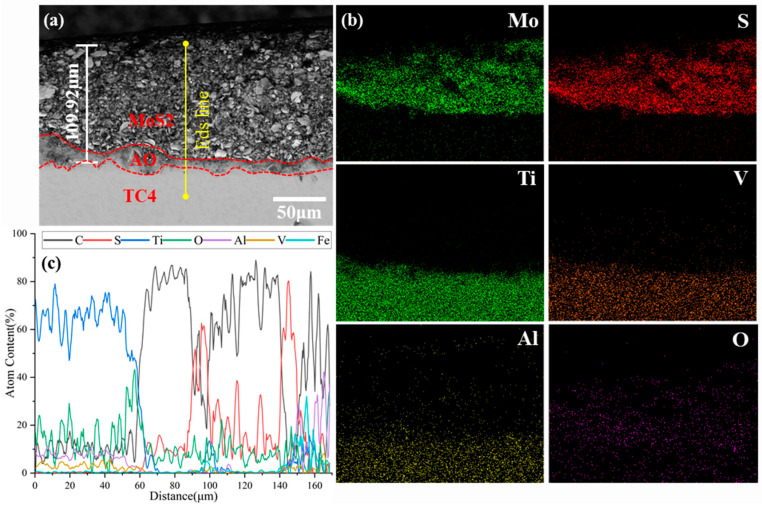
Cross-sectional morphology of TC4+AO+MoS_2_ bolts: (**a**) cross-sectional morphology, (**b**) EDS mapping profiles, (**c**) EDS line scanning.

**Figure 4 materials-18-00123-f004:**
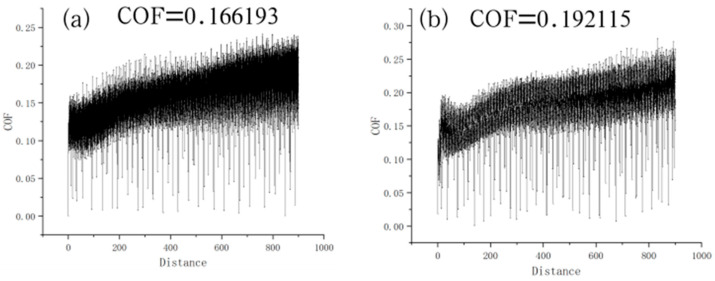
Friction curves of (**a**) TC4+MoS_2_ bolts and (**b**) TC4+AO+MoS_2_ bolts.

**Figure 5 materials-18-00123-f005:**
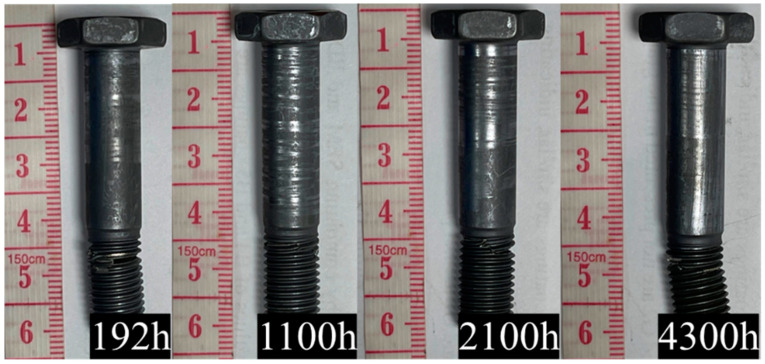
Macroscopic morphology of TC4+MoS_2_ bolts with different corrosion times.

**Figure 6 materials-18-00123-f006:**
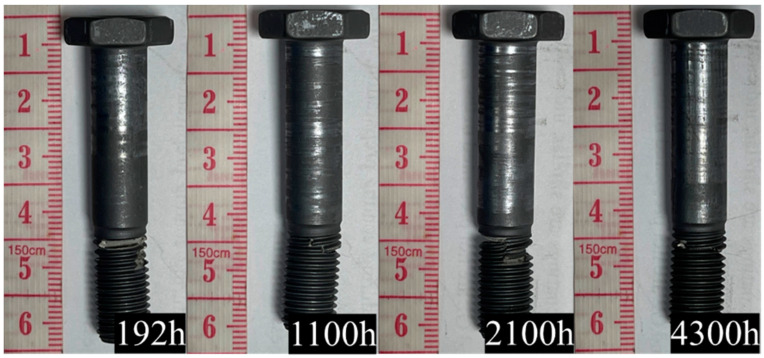
Macroscopic morphology of TC4+AO+MoS_2_ bolts with different corrosion times.

**Figure 7 materials-18-00123-f007:**
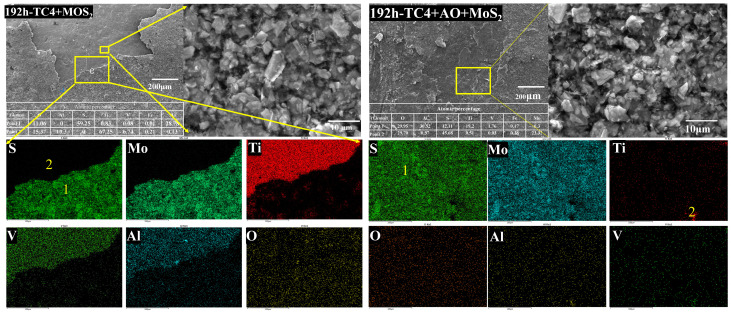
Microscopic morphology and elemental distribution of two bolts after 192 h of salt spray corrosion.

**Figure 8 materials-18-00123-f008:**
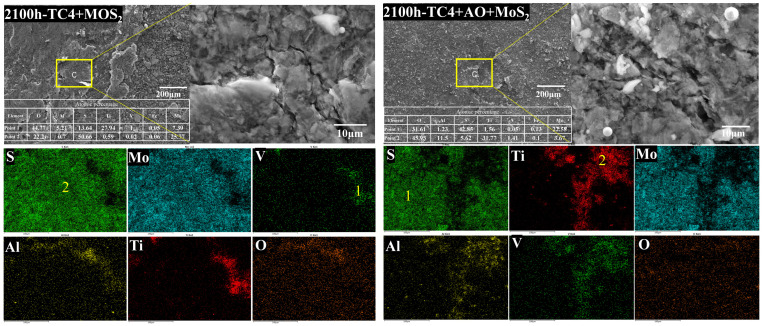
Microscopic morphology and elemental distribution of two bolts after 2100 h of salt spray corrosion.

**Figure 9 materials-18-00123-f009:**
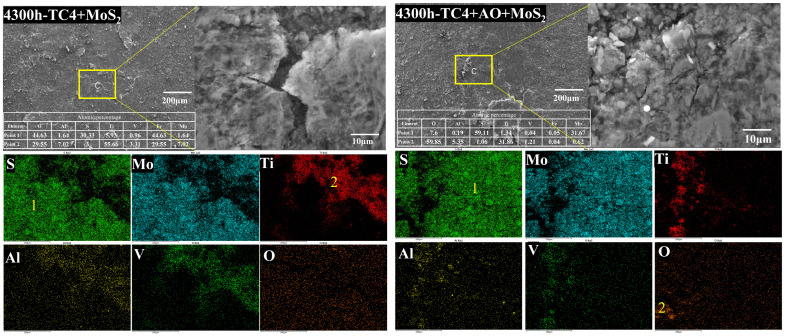
Microscopic morphology and elemental distribution of two bolts after 4300 h of salt spray corrosion.

**Figure 10 materials-18-00123-f010:**
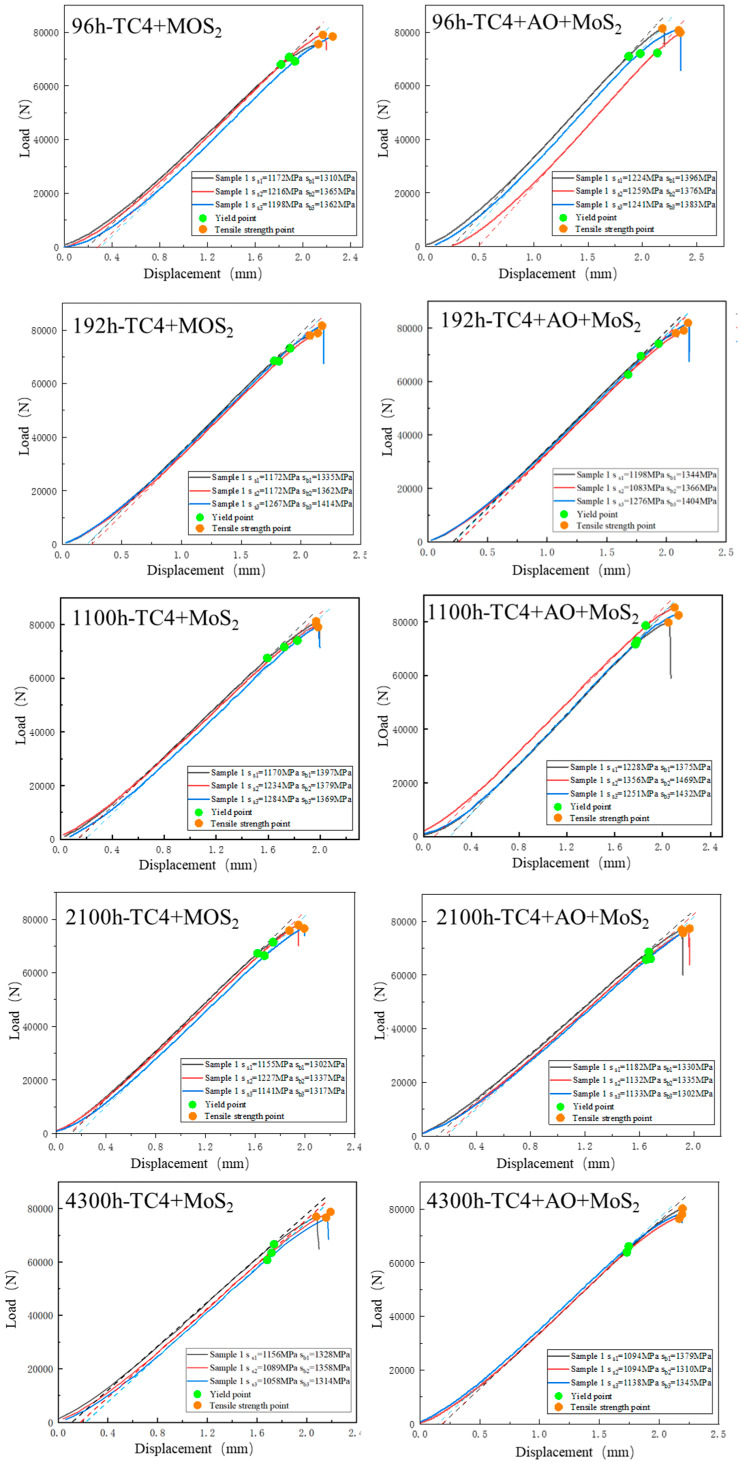
Tensile curves of TC4+MoS_2_ and TC4+AO+MoS_2_ bolts after different times of salt spray corrosion.

**Figure 11 materials-18-00123-f011:**
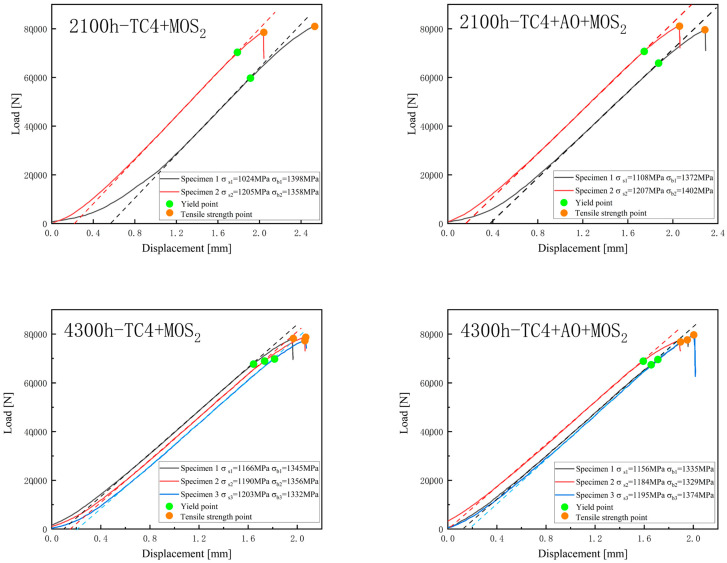
Tensile curves of TC4+MoS_2_ and TC4+AO+MoS_2_ bolts after different times of coastal hanging corrosion test.

**Figure 12 materials-18-00123-f012:**
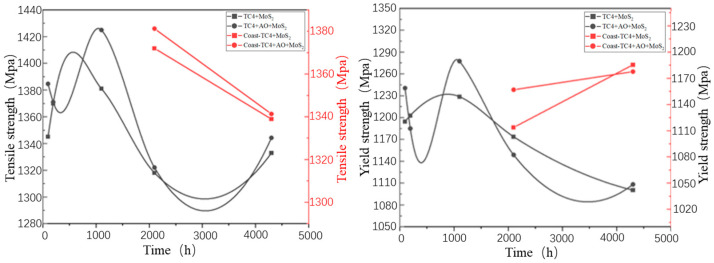
Yield strength and tensile strength of TC4+MoS_2_ and TC4+AO+MoS_2_ bolts after different times of salt spray corrosion and coastal hanging corrosion.

**Figure 13 materials-18-00123-f013:**
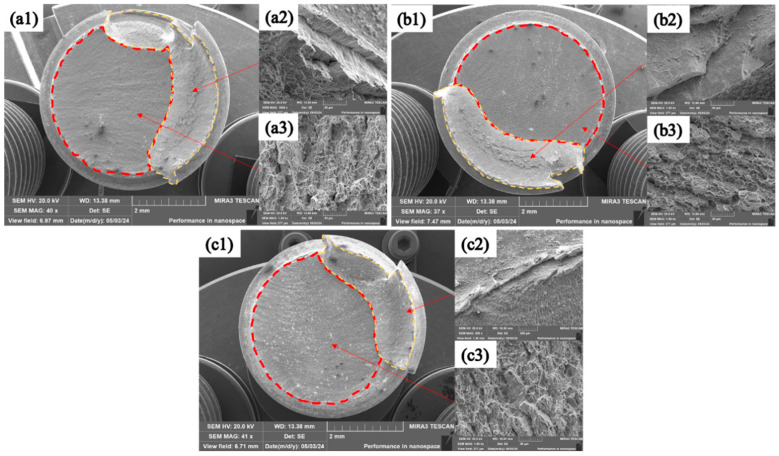
Fracture morphologies of TC4+MoS_2_ bolts: (**a1**–**a3**) after 192 h of salt spray corrosion; (**b1**–**b3**) after 2100 h of salt spray corrosion; and (**c1**–**c3**) after 4300 h of salt spray corrosion.

**Figure 14 materials-18-00123-f014:**
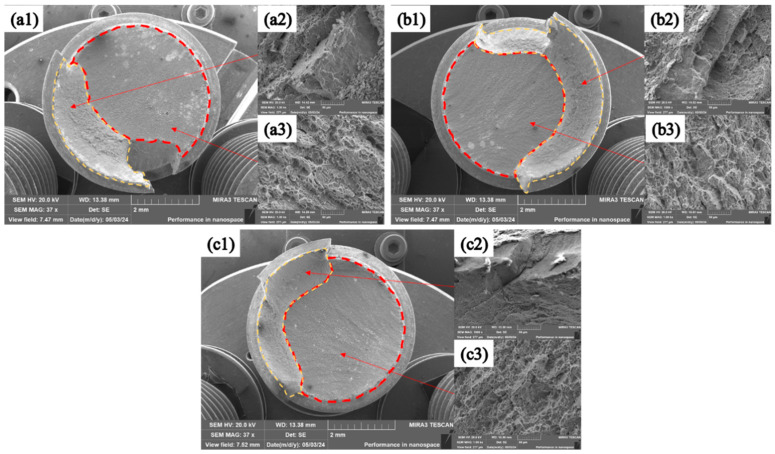
Fracture morphologies of TC4+ac+MoS_2_ bolts: (**a1**–**a3**) after 192 h of salt spray corrosion; (**b1**–**b3**) after 2100 h of salt spray corrosion; and (**c1**–**c3**) after 4300 h of salt spray corrosion.

**Table 1 materials-18-00123-t001:** Yield strength of TC4 bolts treated differently at different times.

	96 h	192 h	1100 h	2100 h	4300 h
TC4+MoS_2_ (salt spray)	1195.3 ± 20.7	1203.7 ± 63.3	1185.7 ± 54.7	1174.3 ± 52.7	1101.0 ± 55.0
TC4+AO+MoS_2_ (salt spray)	1241.3 ± 17.7	1185.7 ± 90.3	1229.3 ± 77.7	1149.0 ± 33.0	1108.7 ± 29.3
TC4+MoS_2_ (costal hanging)	-	-	-	1114.5 ± 90.5	1186.3 ± 20.3
TC4+AO+MoS_2_ (costal hanging)	-	-	-	1157.5 ± 49.5	1178.3 ± 22.3

**Table 2 materials-18-00123-t002:** Tensile strength of TC4 bolts treated differently at different times.

	96 h	192h	1100 h	2100 h	4300 h
TC4+MoS_2_ (salt spray)	1345.7 ± 25.7	1370.3 ± 43.7	1381.7 ± 15.3	1318.7 ± 18.3	1333.3 ± 24.7
TC4+AO+MoS_2_ (salt spray)	1385.0 ± 11.0	1371.3 ± 32.7	1424.94 ± 50.3	1322.3 ± 20.3	1344.7 ± 34.7
TC4+MoS_2_ (costal hanging)	-	-	-	1372.0 ± 20.0	1344.3 ± 12.3
TC4+AO+MoS_2_ (costal hanging)	-	-	-	1387.0 ± 15.0	1346.0 ± 28.0

## Data Availability

The original contributions presented in this study are included in the article. Further inquiries can be directed to the corresponding author.

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
