# Peer review of "The Evolution of the Tensile Properties of MoS2-Coated Titanium Alloy Bolts Under the Synergistic Damage of NaCl Corrosion and Preloading"

_materials, 2024, doi:10.3390/ma18010123_

Round 1
Reviewer 1 Report (Previous Reviewer 1)
Comments and Suggestions for Authors
There are questions still unanswered or partially answered. See below.
I suggest to answer properly to all the revisions.
A2 (Effect of Peeling on Fatigue Life):
Why Unsuitable: The response sidesteps the question by stating that fatigue life data is too dispersed to obtain accurate values. However, even partial observations or trends could provide insights.
Suggestion: Provide any observed trends or qualitative descriptions of how peeling might affect fatigue life. Acknowledge the limitations while detailing what can be concluded.
1.1 (Chemical and Tribological Mechanisms of MoS2):
Why Unsuitable: The reply merely states that the friction behavior of MoS2 coating needs further exploration. This leaves the question unanswered.
Suggestion: Incorporate existing knowledge on MoS2’s low-friction behavior in other environments and hypothesize its relevance to the marine environment.
1.6 (Integration of Recent Technological Developments):
Why Unsuitable: The response acknowledges the addition of literature but does not integrate or discuss how recent advancements like nanoparticles [Targeted dielectric coating of silver nanoparticles with silica to manipulate optical properties for metasurface applications, Materials Chemistry and Physics, 126250, 2022] or multi-functional structures [Multifunctional composites: A metamaterial perspective. Multifunctional Materials, 2(4), 043001, 2019] relate to or enhance the study’s findings.
Suggestion: Discuss the applicability of these technologies to MoS2 and anodized coatings, provide examples or hypotheses on their potential impact, and link them to the study’s objectives or challenges.
3.3 (Uniformity and Stability of Coatings):
Why Unsuitable: The response admits this is an important issue for future study but fails to provide any interim observations or data.
Suggestion: Detail any anecdotal evidence or describe methodologies planned for future studies. Include existing data if available.
4.4 (Dynamic or Cyclic Loading Effects):
Why Unsuitable: The response defers this entirely to future investigation.
Suggestion: Offer hypothetical insights based on related studies or limited experimental data. Discuss the potential implications of cyclic loading based on MoS2's known properties.
4.5 (Cost-Benefit Analysis):
Why Unsuitable: Merely stating that cost-benefit trade-offs will be explored in the future does not contribute to the current discussion.
Suggestion: Provide an initial qualitative assessment or comparative costs of MoS2 with traditional coatings.
Author Response
Dear reviewers,
Thank you very much for your suggestions. We have replied and revised these suggestions one by one, and put them in the following documents in the form of questions and answers. Thank you again for your suggestions.

Reviewer 2 Report (Previous Reviewer 3)
Comments and Suggestions for Authors
In the submitted paper, the authors address the topic of new coatings used to prevent corrosion, specifically MoS2 coating, which is a newly developed method to prevent corrosion and seizure of bolts used in equipment in marine environments.
As stated in the paper, the protection varies depending on the duration of corrosion exposure and the preload force. It is also pointed out that examining the changes in tensile properties and the condition of the coating is of great importance for the maintenance of coated bolts. In the study discussed in the paper, salt spray tests lasting 4300 hours were conducted on titanium bolts coated with MoS2 and subjected to a complex treatment involving anodizing oxidation and MoS2 coating. The authors evaluated the surface coatings of the corroded bolts using SEM and EDS. In the reviewed paper, the coefficient of friction between the coating and the bolt was obtained through reciprocal friction tests and thread friction coefficient measurements.
The results obtained by the authors indicate that the MoS2-coated bolts exhibited lower thread friction coefficients, effectively reducing wear and the risk of seizure after installation. Standard tensile tests were conducted on the corroded bolts for the purposes of the scientific paper. The authors carried out microscopic observations of the corroded surfaces and fracture morphology to determine the changes in the tensile strength of the bolts under the combined effects of corrosion and preload. As shown in the reviewed paper, the MoS2 coating significantly improved the corrosion resistance of the bolts, demonstrating exceptional protective performance, with only minor peeling caused by oxidation-induced cracking during the long 4300-hour salt spray test. As indicated in the manuscript, only a slight decrease in tensile strength was observed with increasing salt spray duration, while the overall tensile strength of the bolts remained above 1300 MPa. The 4300-hour coastal corrosion tests also confirmed the high retained tensile strength.
Based on the results obtained by the authors and presented in the paper, it can be concluded that the coating provides excellent corrosion resistance, having only a minor impact on the tensile strength and fracture characteristics of the bolts under the synergistic effects of marine corrosion and preload.
The obtained results can undoubtedly be applied to solve engineering problems, especially industrial ones. However, before publication, the paper requires several changes and revisions. Here are my detailed comments:
The paper has been resubmitted for the publication process. While the authors have addressed some of the comments, not all have been incorporated. I kindly request the authors to address all the feedback provided. Please add units to the graphs and tables, prepare the graphs in a vector graphic format, and include nomenclature in the paper. Below, I have included my previous comments in full. I maintain my initial review.
The abstract is too long—it should be shortened, and it should not include the results or descriptions of the research methods. The abstract should encourage the reader to read the paper and indicate its topic—maximum 5 or 6 sentences.
The introduction is short but to the point. Congratulations to the authors in this regard. However, I would lean toward making some changes and expanding the introduction, perhaps illustrating the referenced results with the authors' own calculations—I suggest the authors consider this.
For scientific papers, the word "work" should not be used—I recommend replacing it with words like "study," "paper," "manuscript," "article," "scientific article," etc. Please review the paper in this regard, although it seems that no significant corrections are needed, I mention it as an option to consider.
For the experimental material used in the laboratory tests, the word "sample" should not be used; instead, following the ISO, BS, and ASTM standards, the word "specimen" should be used. Please correct the paper accordingly.
I recommend that all graphs be prepared and inserted into the paper in vector graphic format—they can be exported from mathematical applications to the appropriate format, which is vector format. The graphs should preferably be prepared in a software environment such as GRAPHER rather than MS Excel. They will be more aesthetically pleasing and more "engineering-like." I recommend improvements in this regard.
I suggest that all physical units on the drawings and in the tables be written in square brackets, not round brackets.
The paper should necessarily include information about the geometry of the specimens used in the study during the experimental research. I strongly urge the inclusion of a full technical drawing of the specimens used in the tests, along with dimensions and surface roughness annotations. Please add a full technical drawing of the tested bolts with all dimensions and roughness.
Regarding the experimental research, for each type of specimen (bolt), the exact number of specimens used in the experimental tests should be indicated—this applies to all tests conducted in different conditions. It should be specified which signals were recorded during the tests, how they were used, what parameters were determined from them, and what formulas were used to convert the measured quantities.
More distinct example plots of force as a function of displacement should be presented in the paper. All experimental curves should be shown in the figures, and the values derived from Figures 10 and 11 from the tensile tests should be placed in tables. A full statistical analysis should be performed for the obtained values—any obtained data should be subjected to statistical analysis (minimum, maximum, mean, median, range, standard deviation) and displayed in a table along with appropriate explanations and labels. The text of the manuscript should dedicate at least one paragraph to discussing them.
For Figures 10 and 11, I strongly recommend preparing them as vector graphics, and after explaining the quantities, please provide their labels and units in square brackets. For the curves shown in Figure 12, please draw a potential trend line—it begs for the presentation of a regression curve (fairly simple) and the determination coefficient so that a specific physical quantity can be determined for other input parameters. This will enrich the paper and improve its quality.
I recommend adding a nomenclature section to the paper—a complete list of abbreviations, labels, and symbols. The nomenclature can be placed either at the beginning of the paper or at the end, after the bibliography.
The literature in the paper is acceptable and does not raise any objections.
The conclusions should be expanded—potential future research steps should be indicated, as well as how the obtained results can be applied to solve real-world engineering problems, preferably industrial ones, and whether the obtained results could have industrial applications.
The paper has potential, but it must be thoroughly revised—I suggest major revisions.
Please make the necessary corrections to the paper and submit it for further review.
Author Response
Dear reviewers,
Thank you very much for your suggestions. We have replied and revised these suggestions one by one, and put them in the following documents in the form of questions and answers. Thank you again for your suggestions.

Reviewer 3 Report (New Reviewer)
Comments and Suggestions for Authors
Overall, the manuscript presents a comprehensive study that provides valuable insights into the long-term performance of MoS2-coated TC4 bolts in corrosive environments. Authors have conducted a big work on the revision of the paper after peer-review process. However, still some suggested improvements could further enhance the depth and impact of the research.
· It is recommended that the dimensional marker in Figure 2, panel b, be enlarged.
· Additionally, further details should be provided regarding the MoS2 coating process.
· In Figure 12, the trend lines should be replaced with linear lines, as the non-linear relationship does not reflect real behaviour. Alternatively, they could be removed altogether.
· A more detailed discussion of the physical mechanisms responsible for the protective mechanism arising from MoS2 coating would be beneficial for potential readers.
Author Response
Dear reviewers,
Thank you very much for your suggestions. We have replied and revised these suggestions one by one, and put them in the following documents in the form of questions and answers. Thank you again for your suggestions.

Round 2
Reviewer 2 Report (Previous Reviewer 3)
Comments and Suggestions for Authors
The author has addressed almost all of my previous comments in the resubmitted version of the paper.
As a result, the paper has become more valuable and readable, likely to capture the interest of readers.
Additionally, they supplemented the paper with some information.
Overall, the paper is engaging, and I have no further substantive comments to make.
I recommend the paper for publication.
This manuscript is a resubmission of an earlier submission. The following is a list of the peer review reports and author responses from that submission.
Round 1
Reviewer 1 Report
Comments and Suggestions for Authors
Although the research paper shows an experimental design and presentation quality overall; there are some areas that could be enhanced to increase the scientific rigor and depth of the study further. One significant aspect that needs improvement is the lack of quantitative data integration in the conclusions and broader discussions; incorporating more numerical findings could bolster the arguments with clearer insights rooted in data analysis on how well the coatings perform. The research would greatly benefit from a thorough examination of long-term failure mechanisms, like fatigue and brittle fractures – particularly when exposed to extended periods of corrosive and mechanical stress conditions. Furthermore, the lack of control samples or comparison with traditional coatings makes it challenging to understand how much MoS2 and anodizing improve performance. It would be helpful to discuss how the coating process can be optimized and how factors like coating thickness and anodizing parameters affect the outcome. Moreover, discussing implications such, as cost analysis, life-cycle impact and real-world applications will add depth to the research findings.
Below authors can find a detailed section-by section report. I strongly suggest the authors to answer to all the questions raised by the reviewer and insert all the answers properly in the final manuscript.
Abstract
The study could better showcase its depth by including specific numerical data on the reduction in tensile strength and the peeling of the coating layers, for a comprehensive analysis. It would enhance the abstract to clearly distinguish between the effectiveness of the two coating methods ( Mo S 22and anodized + Mo S 22) highlighting their practical implications.
If the abstract doesn't allow room to address the questions mentioned above adequately​ you can include your responses, in the other sections of the manuscript.
A1) What are the exact numerical values of the reduction in tensile strength over the course of the 4300-hour salt spray exposure, and how does this compare to uncoated or traditionally coated bolts?
A2) To what extent does the peeling of the MoS2 coating affect the fatigue life of the bolts under cyclic loading in addition to static tensile tests?
A3) What are the key mechanisms driving the oxidation and peeling of the MoS2 coating in a marine environment, and how do these mechanisms vary between the MoS2-only and anodized bolts?
A4) How does the presence of preload specifically accelerate or mitigate the corrosive effects observed in the salt spray tests, and are there threshold preload values beyond which degradation accelerates?
A5) What is the comparative performance of bolts with MoS2-only coating versus bolts treated with anodic oxidation and MoS2 in terms of long-term corrosion resistance, wear resistance, and mechanical strength?
1. Introduction
The section might be improved by delving into the inner workings involved—especially regarding how the coating and substrate interact chemically and mechanically in corrosive environments. Moreover, it could establish solid links between scientific investigation and tangible real world uses to enhance the study’s significance and practicality. Include comparative information about MoS2 and alternative coatings along, with a more thorough explanation of the anodic oxidation process.
1.1) What specific chemical and tribological mechanisms allow MoS2 to maintain its low friction coefficient and corrosion resistance in aggressive marine environments over long periods?
1.2) How does the microstructure of the anodized layer enhance the adhesion of the MoS2 coating, and what are the key microstructural changes introduced by the anodic oxidation process?
1.3) What are the quantitative performance differences between MoS2 coatings and other traditional coatings (e.g., fluorocarbon or Dacromet) in terms of friction reduction, corrosion resistance, and mechanical degradation under preload?
1.4) What are the predominant failure mechanisms in MoS2 coatings when exposed to combined preload and salt spray corrosion, and how does anodic oxidation mitigate these mechanisms at the microstructural level?
1.5) How do the specific conditions of preload (e.g., magnitude, direction, and distribution) influence the corrosion and mechanical degradation of MoS2-coated bolts, and what are the thresholds for preload where failure rates significantly increase?
1.6) Some recent technology developments are missing and should be taken into consideration in this study such as nanoparticles [Targeted dielectric coating of silver nanoparticles with silica to manipulate optical properties for metasurface applications, Materials Chemistry and Physics, 126250, 2022] and multi-functional structures [Multifunctional composites: A metamaterial perspective. Multifunctional Materials, 2(4), 043001, 2019].
2. Materials and Methods
The section must be improved with detailed explanations regarding the test conditions. Focusing on the environmental variables during corrosion testing and the specific configuration of the friction tests. Present a rationale for factors, like preload and incorporate control samples while elaborating on the techniques used for microstructural analysis.
2.1) What specific environmental conditions (temperature, humidity, and spray distribution) were used during the salt spray corrosion tests, and how do they compare to real-world marine conditions?
2.2) How does the surface roughness of the bolts prior to testing influence the friction and wear properties observed during the friction tests? Was this factor controlled?
2.3) Why was the specific preload value of 40 kN chosen for the tensile and friction tests, and how does this value relate to the typical service conditions that these bolts would experience in aerospace or marine applications?
2.4) What are the comparative friction coefficients for uncoated, traditionally coated, and MoS2-coated bolts, and how do these differences impact the expected lifetime of the bolts in service?
2.5) How did the imaging parameters (e.g., magnification, accelerating voltage) used in the SEM-EDS analysis influence the resolution and accuracy of the microstructural characterization of the coatings?
3. Results and discussion
The section should be improved by conducting in depth quantitative studies that link the level of rust formation with friction levels and the peeling off of coatings along with the physical characteristics such as strength under tension and durability over time under repeated stress cycles in practical settings Also it would be beneficial for the section to delve into potential causes of breakdown in a more thorough manner especially those that may occur following extended use, in real life scenarios.
3.1) What is the quantitative relationship between the degree of corrosion and the observed reduction in tensile strength, and how does this vary between MoS2-only and anodic oxidation-treated bolts?
3.2) How does the surface roughness introduced by anodic oxidation specifically affect the wear resistance and fatigue life of the bolts under cyclic loading conditions?
3.3) What methods were used to assess the uniformity and stability of the MoS2 coatings after different exposure times, and how do these measurements correlate with coating delamination and bolt performance?
3.4) To what extent does the higher friction coefficient observed in anodic oxidation-treated bolts influence the overall durability and lifespan of these components in real-world applications?
3.5) How do the fracture mechanics observed in the tested bolts differ between early and late-stage corrosion exposure, and what role does environmental cracking play in these failure modes?
4. Conclusions
This part could use quantitative results to back up the statements and explore deeper into what causes failures in the long run while comparing it with other coating methods as well as offering ideas for future studies or improvements to make the findings more relevant, to real world engineering problems.
4.1) How does the performance of MoS2-coated and anodized bolts compare quantitatively to other traditional coatings used in marine and aerospace environments in terms of tensile strength retention, friction reduction, and corrosion resistance?
4.2) What are the potential long-term failure mechanisms that might arise after even longer exposure to corrosive environments (beyond 4300 hours), and how might these influence the mechanical performance of the coated bolts?
4.3) To what extent can the MoS2 and anodic oxidation processes be optimized (e.g., adjusting coating thickness, anodizing voltage) to further improve the mechanical properties and corrosion resistance of the bolts?
4.4) How does the presence of dynamic or cyclic loading, in addition to preload, affect the corrosion resistance and mechanical stability of the MoS2-coated and anodized bolts over time?
4.5) What are the potential cost-benefit trade-offs of using MoS2 coatings with anodic oxidation in real-world applications compared to more traditional coatings, and how might these influence decisions regarding material selection for fasteners?
Reviewer 2 Report
Comments and Suggestions for Authors
A review of the article entitled "Evolution of Tensile Properties of MoS2-Coated Titanium Alloy Bolts Under the Synergistic Damage of NaCl Corrosion and Preload" was conducted. Although the paper provides a scientific contribution, there are points that need to be improved to align the study better with the proposed title and focus, and to ensure it can be published. The following considerations should be taken into account:
1. It is recommended that the authors define the term TC4 in the introduction, as it is not widely known outside specific areas and is not mentioned in the title. This would help make the article more accessible to a broader audience.
2. The introduction extensively discusses the friction reduction properties and anti-loosening performance of MoS2-coated bolts. However, the article's title focuses on the evolution of tensile properties, which should receive more emphasis.
3. The experimental procedure lacks a clear and detailed explanation of how the evolution of tensile properties will be evaluated over the corrosion exposure periods (96, 192, 1100, 2100, and 4300 hours). This point should be elaborated upon to better align the study with the main objective.
4. The description of the microstructural characterization tests could be improved by including whether surface preparation or chemical etching was performed, as well as the parameters used in microscopy.
5. The color standardization in the sulfur mapping and in the EDS scanning should be adjusted to avoid confusion. For instance, sulfur appears in different colors in the same image (green and red). Additionally, the behavior of carbon and sulfur observed at the interface between the Ti-rich substrate and the coating layer needs to be explained. There is a lack of adequate discussion regarding the Fe atomicity, which appears higher compared to Al and V, without explanation in the text.
6. There are inconsistencies in the figure descriptions, such as the mention of Figure 3, which is not cited in the text.
7. Figure 4 describes the friction coefficient, correct? Why is the y-axis labeled COF while the graph highlights COF = 0.166193, and in the text, this value is indicated as the maximum point? The graph shows points up to nearly 0.25, which is very confusing. Additionally, the noise level is relatively high. Other studies presenting friction coefficients for TC4 show better behavior (10.1002/adem.202300517; https://doi.org/10.1080/00150193.2021.1905720, for example).
8. Did the authors develop equations 2.1 and 2.2, or are they following previous work?
9. First, the authors mention that “Figure 10 shows the tension curves of two different types of treated bolts after salt spray corrosion.” Then, they state that “Figure 10 and 11 show tensile curves of TC4+MoS2 and TC4+AO+MoS2 bolts after different times of coastal hanging corrosion, respectively.” The authors should standardize the text to make it more scientific and clear; currently, it is confusing.
10. Although the text mentions that the experiments were conducted in triplicate, no error bars were presented in any graph or throughout the text, which compromises the clarity and reliability of the data presented.
11. The discussion on the "evolution" of mechanical properties over time is superficial. Given that the title suggests a dynamic monitoring of this evolution, it would be beneficial to include a more detailed analysis of the degradation rate of tensile properties over the corrosion periods. Connecting this discussion to the term "evolution" in the title would reinforce the study's depth.
12. On page 4, line 135, 42.55?? What unit of measurement is being used?
13. The conclusion mentions preload but does not discuss in detail how the combination of corrosion and preload results in the synergistic damage mentioned in the title. It would be helpful to expand this section to better explain how these two factors interact and impact the integrity of the bolts over time.
Comments on the Quality of English LanguageThe text presents several typographical errors and inconsistencies that compromise its clarity. A thorough and careful revision is necessary to ensure terminological accuracy and alignment between figures, graphs, and descriptions, as well as adjustments to improve readability and scientific rigor.
Reviewer 3 Report
Comments and Suggestions for Authors
In the submitted paper, the authors address the topic of new coatings used to prevent corrosion, specifically MoS2 coating, which is a newly developed method to prevent corrosion and seizure of bolts used in equipment in marine environments.
As stated in the paper, the protection varies depending on the duration of corrosion exposure and the preload force. It is also pointed out that examining the changes in tensile properties and the condition of the coating is of great importance for the maintenance of coated bolts. In the study discussed in the paper, salt spray tests lasting 4300 hours were conducted on titanium bolts coated with MoS2 and subjected to a complex treatment involving anodizing oxidation and MoS2 coating. The authors evaluated the surface coatings of the corroded bolts using SEM and EDS. In the reviewed paper, the coefficient of friction between the coating and the bolt was obtained through reciprocal friction tests and thread friction coefficient measurements.
The results obtained by the authors indicate that the MoS2-coated bolts exhibited lower thread friction coefficients, effectively reducing wear and the risk of seizure after installation. Standard tensile tests were conducted on the corroded bolts for the purposes of the scientific paper. The authors carried out microscopic observations of the corroded surfaces and fracture morphology to determine the changes in the tensile strength of the bolts under the combined effects of corrosion and preload. As shown in the reviewed paper, the MoS2 coating significantly improved the corrosion resistance of the bolts, demonstrating exceptional protective performance, with only minor peeling caused by oxidation-induced cracking during the long 4300-hour salt spray test. As indicated in the manuscript, only a slight decrease in tensile strength was observed with increasing salt spray duration, while the overall tensile strength of the bolts remained above 1300 MPa. The 4300-hour coastal corrosion tests also confirmed the high retained tensile strength.
Based on the results obtained by the authors and presented in the paper, it can be concluded that the coating provides excellent corrosion resistance, having only a minor impact on the tensile strength and fracture characteristics of the bolts under the synergistic effects of marine corrosion and preload.
The obtained results can undoubtedly be applied to solve engineering problems, especially industrial ones. However, before publication, the paper requires several changes and revisions. Here are my detailed comments:
The abstract is too long—it should be shortened, and it should not include the results or descriptions of the research methods. The abstract should encourage the reader to read the paper and indicate its topic—maximum 5 or 6 sentences.
The introduction is short but to the point. Congratulations to the authors in this regard. However, I would lean toward making some changes and expanding the introduction, perhaps illustrating the referenced results with the authors' own calculations—I suggest the authors consider this.
For scientific papers, the word "work" should not be used—I recommend replacing it with words like "study," "paper," "manuscript," "article," "scientific article," etc. Please review the paper in this regard, although it seems that no significant corrections are needed, I mention it as an option to consider.
For the experimental material used in the laboratory tests, the word "sample" should not be used; instead, following the ISO, BS, and ASTM standards, the word "specimen" should be used. Please correct the paper accordingly.
I recommend that all graphs be prepared and inserted into the paper in vector graphic format—they can be exported from mathematical applications to the appropriate format, which is vector format. The graphs should preferably be prepared in a software environment such as GRAPHER rather than MS Excel. They will be more aesthetically pleasing and more "engineering-like." I recommend improvements in this regard.
I suggest that all physical units on the drawings and in the tables be written in square brackets, not round brackets.
The paper should necessarily include information about the geometry of the specimens used in the study during the experimental research. I strongly urge the inclusion of a full technical drawing of the specimens used in the tests, along with dimensions and surface roughness annotations. Please add a full technical drawing of the tested bolts with all dimensions and roughness.
Regarding the experimental research, for each type of specimen (bolt), the exact number of specimens used in the experimental tests should be indicated—this applies to all tests conducted in different conditions. It should be specified which signals were recorded during the tests, how they were used, what parameters were determined from them, and what formulas were used to convert the measured quantities.
More distinct example plots of force as a function of displacement should be presented in the paper. All experimental curves should be shown in the figures, and the values derived from Figures 10 and 11 from the tensile tests should be placed in tables. A full statistical analysis should be performed for the obtained values—any obtained data should be subjected to statistical analysis (minimum, maximum, mean, median, range, standard deviation) and displayed in a table along with appropriate explanations and labels. The text of the manuscript should dedicate at least one paragraph to discussing them.
For Figures 10 and 11, I strongly recommend preparing them as vector graphics, and after explaining the quantities, please provide their labels and units in square brackets. For the curves shown in Figure 12, please draw a potential trend line—it begs for the presentation of a regression curve (fairly simple) and the determination coefficient so that a specific physical quantity can be determined for other input parameters. This will enrich the paper and improve its quality.
I recommend adding a nomenclature section to the paper—a complete list of abbreviations, labels, and symbols. The nomenclature can be placed either at the beginning of the paper or at the end, after the bibliography.
The literature in the paper is acceptable and does not raise any objections.
The conclusions should be expanded—potential future research steps should be indicated, as well as how the obtained results can be applied to solve real-world engineering problems, preferably industrial ones, and whether the obtained results could have industrial applications.
The paper has potential, but it must be thoroughly revised—I suggest major revisions.
Please make the necessary corrections to the paper and submit it for further review.
Comments on the Quality of English LanguageMinor editing of English language required.